# The Relationship between Childcare-Giving Arrangements and Children’s Malnutrition Status in South Africa

**DOI:** 10.3390/ijerph20032572

**Published:** 2023-01-31

**Authors:** Matshidiso Sello, Sunday A. Adedini, Clifford Odimegwu, Rebaone Petlele, Talent Tapera

**Affiliations:** 1Demography and Population Studies Programme, Schools of Public Health and Social Sciences, University of the Witwatersrand, Johannesburg 2000, South Africa; 2Centre for Social Development in Africa, University of Johannesburg, Johannesburg 2092, South Africa; 3Department of Demography and Social Statistics, Faculty of Social Sciences, Federal University Oye-Ekiti, Oye-Ekiti 371104, Nigeria

**Keywords:** childcare, secondary care, multiple childcare, malnutrition, stunting, overweight, crèches, South Africa

## Abstract

Multiple caregiving arrangements have become common for childcare globally, and South Africa is no exception. Previous childcare studies mainly focused on the caregiver and household characteristics. Evidence on the influence of childcare on malnutrition is sparse. This study aimed to examine the relationship between exposure to secondary and multiple forms of care and child malnutrition, with a particular focus on child stunting and overweight among children. A cross-sectional study of a sample of 2966 dyads of mothers and children under five were analysed from the 2017 National Income Dynamics Study (NIDS) Wave 5. Descriptive and inferential statistics were used to analyse the data. The results indicated that 22.16% of the children were stunted and that 16.40% were overweight. Most children were mainly cared for at home (67.16%) during the day. Some results of the obtained multivariable analyses show that lack of being cared for in a crèche or school during the day was significantly associated with stunting (odds ratio (OR) 2; confidence interval (CI) 1.10–3.62, *p* < 0.05) and overweight (OR) 3.82; (CI) 1.60–9.08, *p* < 0.05). Furthermore, in this study, 69.88% of children who were cared for at home by the primary caregiver had no other forms of multiple care arrangements. The results showing high stunting and overweight rates among children cared for at home suggest that the government needs to look into supporting caregiver parenting. The high unemployment rates in the country highlight the importance of socioeconomic status in childcare and its implication for children’s nutritional outcomes. The study’s findings suggest the need for innovative strategies to address the challenges associated with multi-caregiving which negatively affects children’s nutritional outcomes.

## 1. Introduction

Malnutrition among children under five remains a persistent global problem, particularly in low- and middle-income countries (LMICs). Malnutrition is defined as a health condition resulting from either excessive or deficient nutrients. It is therefore grouped into two broad categories: undernutrition and overnutrition [1]. Sub-forms of undernutrition include stunting (low height-for-age), wasting (low weight-for-height), and underweight (low weight-for-age). Overnutrition refers to overweight (high weight- for-age) and obesity (high weight-for-height) [2]. In Africa, the concurrence of undernutrition (characterised by stunting) and over nutrition (characterised by overweight) is not uncommon [3]. Stunting has been a pressing development challenge, with nearly 56.6 million children being stunted [3]. East African countries, such as Burundi (57.7%) and Malawi (47.7%), have the highest prevalence of stunting [4]. The prevalence of overweight in children under the age of five in middle-income countries was nearly 79% [5]. However, in Southern and Northern Africa, overweight children under five ranged between 10.6% and 13%, with the exception of countries such as Libya, where 25.4% of children under five years old were overweight in 2020 [6]. Since 2000, Africa has seen nearly a 24% increase in the percentage of children under five who are overweight [7,8].

In South Africa, child stunting and child overweight are the leading indicators of child malnutrition, with 27% of children under five still stunted and 13% overweight. These indicators are especially important given that, despite the nutrition interventions that the government has implemented, such as the child support grant [9], stunting and overweight are still staggeringly high. The situation for children has not improved much in almost three decades: 29.7% with stunting and 13.6% overweight in 1990 [3]. Additionally, in 2016, 4% of the children were both stunted and overweight [3]. This overlap is referred to as a double burden of child malnutrition [10]. The sustainable development goals (SDGs), 1: no poverty; 2: zero hunger; and 3: good health and wellbeing, are part of international plans seeking to ensure that children are well cared for [11,12,13,14,15]. The South African government has made efforts to reduce child malnutrition to ensure that SDG goals 1, 2, and 3 are achieved through its National Development Plan (NDP), which seeks to eradicate poverty and all its forms, by making provision for social grants, by introducing the National Policy on Food Security, and by providing accessible and free health care for all children under five [16].

Childcare is a pathway to early childhood development (ECD). It is critical to children’s growth and human capital [17,18]. The nutritional outcomes and the wellbeing of children under five years old depend on the care they receive, access to basic health services, and their household socioeconomic status [10,19]. The type of care that children receive has implications on their health, diet, nutrition, development, and survival. Lack of adequate care for children under five may expose them to adverse health outcomes. These include child malnutrition, which has long-term effects on future learning abilities, earning potential, morbidities, and disease patterns [4,20,21]

In South Africa, like many other low and middle-income countries, extended caregiving of children is common. It provides a foundation for children’s social security. Childcare-giving has become a shared responsibility by biological caregivers and other adults who may or may not be the biological family [22]. Furthermore, the shared responsibility in childcare-giving is due to the changing dynamics of family structures, characterised by single-parent households where a parent raises a child without a partner living with them. These changing dynamics could be due to several factors, including death, divorce or separation, and migration. A study in South Africa found that 60% of South African children have absent fathers [23]. This is defined as fathers who are alive but do not reside with their children, see them infrequently, and/or rarely give them financial support [23]. More than 40% of South African mothers are single parents [24]. At the time of our research, nearly 21.3% of children did not live with their biological caregivers and 32.7% of children lived with both caregivers [25]. Marriage rates declined by 22.5% between 2011 and 2019. Contexts such as migration and orphanhood have led to increased multigenerational family structures. These changes in family structures may contribute to decisions about the best care for the child being negotiated among family members and, in some cases, the state intervening [26,27].

Due to caregivers being employed and often having long hours at work, extended family members, day mothers, and crèches have become the most commonly available options and substitute sources of care for primary caregivers [26]. Childcare can be demanding emotionally and physically given the amount of time that goes into preparing food and nurturing children. Therefore, the environment, such as the home or the alternative environment in which the child is brought up or cared for, is critical in the child’s growth and development [26,28]. Growing evidence from the literature suggests that increasing urbanisation has led to a demand for childcare, which includes formal crèches and informal forms of childcare. The formal crèches are registered with the government, or formally registered as non-profit organisations (NPOs) or nongovernmental organisations (NGOs). They are also regulated by the government, and the regulation includes unannounced inspection visits from environmental health practitioners, which includes checks on sanitation, staff-to-child ratios, separate bathroom facilities and kitchen area, separate sick rooms, and indoor and outdoor space based on the area per child [29]. Informal caretakers who are unregistered crèches, day mothers, and extended family members care for the child outside the home setting [11]. Some studies have indicated that instability in providing childcare threatens the child’s health and nutritional outcomes [30,31].Children are a vulnerable group within society, and their health and wellbeing should be a priority.

Previous studies have focused much of their attention on childcare in the ‘home setting’. This would include a household’s socioeconomic status, living conditions, the number of people living in the home, and the primary caregiver’s characteristics. These include factors such as age, employment status, and level of education [14,19,32,33,34]. For example, if caregivers have a higher level of education, they would be knowledgeable about feeding practices and the nutritional value of different foods. A lower socioeconomic position serves as a significant risk factor impacting the household’s nutrition due to insufficient food consumption [35,36]. In addition, the notion that the immediate primary caregivers are responsible for ensuring that young children’s needs are met assumes that the home is the only place where childcare takes place. This ignores the role of secondary and multiple childcare providers. Evidence of multiple childcare arrangements in and outside the home setting is sparse, primarily because, with working caregivers, childcare may also happen outside the home setting. This study examines the relationship between exposure to secondary and multiple forms of care and child malnutrition, with a particular focus on child stunting and overweight among children.

## 2. Materials and Methods

This section outlines our materials and methods.

### 2.1. Study Design

In this study, an analysis was drawn from the National Income Dynamics Study’s (NIDS) Wave 5 data, collected in 2017. The NIDS is a nationally representative panel study that follows approximately 10,000 households and 28,000 individuals in South Africa [37]. These individuals were followed since 2008 across the whole country. The range of topics collected from the NIDS includes demographics, labour market participation, health status, education, household socioeconomic status, living conditions, and living arrangements. The first sample was enrolled in 2008. To date, five waves have been conducted. This study only analysed the cross-sectional data collected in 2017 of dyads of children under five and their mothers. This was survey data collected face-to-face using tablets loaded with the questionnaire [37]. For the children to be included in the sample, they had to have complete anthropometric figures (height, weight, and mid-upper circumference (MUAC). In the analysis of this study, data were weighted to make inferences about the South African population.

### 2.2. Sample Design

NIDS targeted the population from private homes and occupants in workers’ residences and convents and is not inclusive of other communal housing such as prisons, old age homes, students hostels, and hospitals [37]. The 2017 NIDS Wave 5 data used the stratified, two-stage cluster sample design to select the dwelling units to be visited. The 2003 Master Sample of 3000 Primary Sampling Units (PSUs) of Statistics South Africa was used to create a sample of 400 PSUs in the first phase. Within each PSU, eight non-overlapping samples of ten or twelve dwelling units were methodically drawn to create the Master Sample. The explicit strata in the Master Sample are the 53 district councils (DCs). The sample was proportionally allocated to these 53 strata, and PSUs were selected within strata with probability proportional to size. The sample was not intended to be indicative of province-level population trends [37]. In this analysis, the structure was not considered.

The sample for this study was limited to 2966 weighted dyads of mothers aged 15–49 and their children under five with complete anthropometric data.

### 2.3. Measures

#### 2.3.1. The Outcome Variables: Child Health Outcomes

The outcome variable was malnutrition, as indicated by stunting and overweight children under five years of age, that is, children 0–59 months old.

Children were weighed in kilograms and had their height measured in meters by trained field workers at their homes. The field workers were employed by the South African Labour and Development Research Unit (SALDRU) at the University of Cape Town. Stunting was defined as height-for-age, and overweight was defined as body mass index (BMI) for age. The WHO child growth standards and the WHO AnthroPlus software were used to compute the indicators for child malnutrition [38]. Children with a z-score of ≤−2 were classified as being stunted, and children presenting with a z-score of ≥2 were classified as overweight/obese [38].

#### 2.3.2. Main Independent Variable: Multiple Forms of Childcare

The NIDS data asked questions about who else cared for the child besides the primary caregiver [37]. In this study, the multiple forms of childcare arrangements outside the primary caregiver were classified as in the home setting or outside the home setting. At home, the child may receive care from the other parent, grandparents, and relatives/non-relatives. Outside the home setting, the child may receive care from the crèche or school.

#### 2.3.3. Covariates

This analysis considers that multiple factors may influence children’s nutritional status. This may operate at different levels, such as the individual and household levels [10,39]. These factors may also influence whether the child is exposed to multiple caregiving or not [30]. Previous studies have argued that childcare has become a shared responsibility due to changing family dynamics, household socioeconomic status, and caregiver employment status [26,27,40]. Various data regarding children and their families were gathered as part of the NIDS study, which include changes in income, health and wellbeing, assets, and expenditure [37]. 

#### 2.3.4. Caregiver- and Household-Level Variables

While certain primary caregiver characteristics may influence the multiple forms of childcare and children’s health outcomes, such as stunting, overweight, and underweight [13,41,42], this analysis included the demographic and sociodemographic variables of the mother who resided with the child. This included age, marital status, religion, ethnicity, education, employment status, and employment history (what caregivers were doing a year before the survey was conducted, such as taking care of others, in university, volunteers, retired, et cetera). The child’s household-level variables included household size, access to water, toilet type, and electricity. Some of the child’s individual-level characteristics include gender, age in months, race, disease episode, health care utilisation, and receiving multiple forms of care.

### 2.4. Data Analysis

Stata version 17 was used for data analysis considering the population size of 2966 dyads of mothers and their children, with a 5% margin of error and a 95% confidence level [43]. The main independent variable (multiple forms of childcare) is categorised as the care from the other parent, grandparents, and relatives/nonfamily caregivers and no other carer. The data were analysed at three levels: the univariate, bivariate and multivariable levels. At the univariate level, descriptive statistics of frequency distributions and percentages of independent variables were shown. At the second level, a bivariate analysis was conducted using the chi-square test to establish the strength of the relationships between the independent and dependent variables. At the third level, a multivariable binary logistic regression analysis was conducted. The selection of the independent variables was based on (1) the UNICEF’s Conceptual Framework on Causes of Undernutrition and the Nurturing Care Framework [44,45]; (2) the variables that had a strong association from the chi-square test; and (3) variables that had a strong correlation using the Variance Inflation Factor (VIF), a test that measures the extent of multicollinearity and identifies variables that should not be included simultaneously as independent variables in a multiple (logistic) regression function. Variables that depicted a VIF over 4 or a tolerance below 0.25 indicate that multicollinearity may exist [46], and these variables were excluded from the analysis. Firstly, Model 1 presents a univariate logistic regression, where the outcome variables were modelled with individual covariates. Secondly, Model 2 presents the multivariable binary logistic regression, which included the outcome variables and all the selected covariates, including gender, age, race, ethnicity, medical aid, child support grant, multiple forms of care, and care during the day.

## 3. Results

### 3.1. Caregiver-Level Characteristics

The summary statistics giving descriptive background information on the caregiver are presented in Table 1. A total of 2966 women aged 15–49 were surveyed. The majority were single or divorced (73.87%), had no medical aid (89.98%), had up to a secondary level of education (62.37%), or were unemployed (57.78%), and over half were Sesotho-speaking (51.35%).

### 3.2. Child-Level Characteristics

The child-level characteristics are reported in Table 2. The results indicated that 22.16% of the children were stunted and 16.40% were overweight. Among these children, 5.70% of the same children had concurrent stunting and overweight. In Table 2 below, regarding gender, there was a balance between female (49.86%) and male (50.14%) children of those sampled. The majority were cared for at home during the day (67.19%), received multiple care (66.96%), and received multiple forms of care from other parents (36.08%) and grandparents (21.11%). Likewise, many children were recipients of the child support grant (80.59%) and had no medical aid (93.09%).

Since more caregivers were utilising multiple childcare strategies [14]; Table 3 presents the distribution of children receiving multiple forms of childcare during the day besides care from the primary caregiver. This study found that 65.29% of children were cared for by their grandparents and were cared for at home during the day. Similarly, of the children who were cared for by the main caregiver, 69.88% were cared for at home, 22.69% attended a crèche, and 7.43% were in school.

The chi-square test of independence results show a significant association between the child’s age, stunting, and being overweight *p* < 0.005. Children aged 24–35 months had a higher percentage of being stunted (25.08%), and children 12–23 months had a higher percentage of being overweight (32.69%) compared to other age groups. Where the child is cared for during the day is significantly associated with stunting and overweight (*p* < 0.005). Children cared for at home during the day have the highest percentage of stunting (24.98%) and overweight (18.99%). However, variables such as the child support grant and medical aid were the only variables associated with child stunting, with children who are recipients of the child support grant having the highest percentages, at 23.43% and 22.99% of children with a medical aid being stunted (*p* < 0.005). There was no significant relationship among children receiving multiple forms of care (from other parents, grandparents, and relatives/non-relatives and no other carer), stunting (*p* = 0.768), and being overweight (*p* = 0.539).

As earlier mentioned, prior research demonstrated that child, caregiver, and family characteristics may be related to children being exposed to multiple childcare arrangements [30,40,47]. The study’s child-level variables included sex, age, race, ethnic groups, birthplace, child sickness, access to medical aid, and whether the family received a child support grant.

Table 4 presents the multivariable binary logistic regression model to account for individual child and caregiver-level predictors associated with child malnutrition. The selection of the variables in this table was guided by the reviewed literature, UNICEF Conceptual Framework of Causes of Malnutrition (1990), and the Nurturing Care Framework. Prior to fitting this model, we ran the VIF model. The mean VIF for the variables was 1.51, and the VIF values for the selected variables ranged from 1.09 to 3.39.

In this model, we found that gender, age, ethnicity, and childcare during the day were significantly associated with child stunting. Female children had 17% lower odds of being stunted than male children. Likewise, children cared for at home during the day had 77% higher odds of being stunted than children cared for at crèches or school. Socioeconomic status, such as not having medical aid, also influences child health outcomes; 90% of children without medical aid had higher odds of being stunted and 36% had lower odds of being overweight. Children aged 12–23 months had 2.3 times higher odds of stunting and 7.11 times higher odds of being overweight.

We conducted an additional multivariable binary logistic regression to determine other factors associated with child malnutrition (see Table 5 and Table 6). Table 5 only models the variables that were associated with child stunting and were statistically significant. Similarly, Table 6 models variables that were associated with overweight and were statistically significant. These were also child-level variables, as children under 5 were our group of interest. 

Using a multivariable logistic regression to further examine the key factors for stunting and overweight, we found that age was significantly associated with child stunting. Children aged 12–23 months had higher odds of stunting (OR = 2.58, CI:1.60–4.14). African children had 60% higher odds of being stunted compared to children of other races. Likewise, children with no medical aid had higher odds of being stunted (OR = 2.33, CI: 1.35–4.03). Similarly, children who were recipients of the child support grant had higher odds of being stunted (OR = 1.46, CI: 1.04–2.05). Children cared for at home during the day had twice the odds of being stunted compared with children who went to schools or crèches (CI: 1.10–3.62).

Children 12–23 months had eight times higher odds of being overweight than other age groups (CI: 4.71–14.14). Likewise, children cared for at home had 3.82 times higher odds of being overweight than children at school or crèches (CI: 1.60–0.08).

## 4. Discussion

This study presents the findings from the 2017 Wave 5 NIDS, which tracked changes in the wellbeing of South Africans [37]. This study indicated stunting to have a higher prevalence than overweight. The reported prevalence of stunting and overweight in the current study is indicative of South Africa not having made much progress in reaching the 40% global nutrition target of reducing stunting and halting the epidemic of obesity among children by 2025 [48]. In fact, this study found the stunting and overweight result to resemble the South African National Health and Nutrition Examination Survey (SANHANES) conducted a decade ago, demonstrating the persistence of stunting and overweight among children [49]. We also found significant differences in gender when looking at nutrition outcomes. Girls presented with lower odds of stunting compared to boys. This finding is consistent with another study conducted by Thurstans et al., a meta-analysis on 44 studies, which found stunting to be more prevalent among boys in 32 out of 38 studies [50]. Furthermore, our result nullifies the assumption that girls have a higher disadvantage in nutrition outcomes [51,52]. Age was significantly associated with stunting and overweight, with children in the age groups 12–23 and 24–36 months having the highest odds. A similar study in South Africa has attributed stunting and overweight to be caused by improper dietary supplementation after the breastfeeding period and, in some cases, continued breastfeeding unaccompanied by adequate nutritious foods [53].

The high prevalence of overweight reported in this study resonates with the rising global trends of obesity, which was previously predominantly a high-income countries’ challenge [54]. Additionally, evidence in this study points to children cared for at home having the highest risk of being overweight compared to those in schools or crèches. Some studies highlight the nutritional inadequacy of the food given to children under five in the home setting. These include a diet that is short of fruits and vegetables and includes sugar-sweetened beverages and unhealthy foods such as takeaways with excessive fats and salt [33,55]. In addition, to the diet, some studies have argued that children cared for at home were at a higher risk of obesity due to excessive screen time, parental/caregiver unemployment, poverty, and lack of nutritious foods [56,57].

In South Africa, schools and registered crèches are regulated. The finding of children who are in crèches and in schools presenting with better nutritional outcomes compared to children cared for at home is not surprising. The South African education policy exempts school fees among learners from low-income families [58,59], and there is a provision for the National School Nutrition Programme (NSNP) so that disadvantaged learners receive breakfast and lunch at the school [60,61]. Similarly, in registered crèches, children who are recipients of the child support grant (CSG) and whose caregivers qualify under the means test can benefit from the R17-a-day nutrition grant subsidy [60]. However, the same provision of exempting fee payment for crèches among low-income families is not granted because crèches are privately owned [59]. The study’s finding of over half of the children being cared for at home is evidence of the inequalities and socioeconomic disadvantages that exist among many caregivers in the country [59]. The high unemployment rate has implications in the caregiver’s ability to send children to crèches because of the lack of money to pay for crèche fees. Another study conducted in South Africa has found that the lack of childcare arrangements may restrain caregivers with young children in seeking employment [60]. Moreover, the improved nutritional outcomes associated with children attending registered crèches, studies by Maharaj and Dunn [60] and Yeleswarapu and Nallapu [61] have found that crèche attendance among children comes with added benefits for the child and the caregivers. The benefits for the child include stability, routine, socialisation, interaction, and child supervision. This helps them reach their developmental milestones. Furthermore, for the caregiver, the benefits include peace of mind when a child is in a safe and supervised environment. It also frees up time for the caregiver to seek employment and to attend job interviews [55,56]. In middle-income countries such as South Africa, a no-fee crèche would be a viable intervention, given the countries NDPs which seek to tackle poverty through providing social security [60,61].

Some studies distinguish the type of care received in formal and informal crèches. Informal crèches are often characterised by overcrowding and staffing challenges. A study performed by Atmore et al. has argued that caregivers may have various reasons for choosing by whom or where the child is cared for in their absence. Often, this decision is based on their financial means [62]. Caregivers who cannot afford formal registered crèches will send their children to informal crèches or day mothers. This is a risk factor for children being underweight, stunted, and wasted [32,63,64]. Social support networks are key in providing nurturing and care for children under five. We found that, besides the caregivers, over a quarter of children did not have multiple childcarers. Grandparents and relatives/non-family members assumed the role of providing secondary care in the absence of the primary caregiver. Social networks are also important for caregivers who are employed. In their absence, they require support from others for the care of their children, care work which may be paid or unpaid.

Socioeconomic status plays a critical role in childcare and nutritional outcomes. A study of mothers working in the informal sector reported that, to cut costs of paid care, mothers had to leave their children at home to be cared for by their siblings [65]. We also found that the majority of children were recipients of the CSG. In South Africa, being a recipient of the CSG is indicative of low socioeconomic status. To qualify for the grant, the child’s primary caregiver must receive an income of less than ZAR 48,000 a year. Therefore, high unemployment and being a recipient of the CSG may support the significant result found on child stunting. In a study conducted in South Africa, Kekana et al. found that, while the CSG was intended to enhance the child’s wellbeing, the grant money was used for multiple expenses in the household, such as paying stokvels (in the South African context, a stokvel refers to a club whose objective is savings or investments), food, water, and electricity [66]. This study also shows that children who were not receiving the CSG had 46% higher odds of receiving multiple childcare. This suggests that these children’s caregivers may require additional support in caring for them while the caregivers may be at work. In South Africa, crèches provide an essential service in childcare; however, this sector has not received the much-needed support from the government. This is inclusive of the registered crèches that are recipients of the R17-a-day nutrition grant with which the centres are not able to adequately meet their running expenses [59]. The need for crèches has been well recognised, given that, as of October 2022, there has been a proposal before parliament to make at least two years of early childhood education (creche attendance) be made compulsory before children enter grade 1. However, this proposal has vastly been made from the education and learning perspective of children and not for the nutritional outcomes of the children. Some academics have also supported the notion to make the last two years of ECD compulsory but acknowledge that it should be free, particularly for caregivers in low-income settings [67]. Given that free education is part of the children’s rights and is part of the constitution, the South African government should make crèches tuition-free given the numerous benefits that children receive (nutritional benefits, learning outcomes, and socialisation). Therefore, it is viable for middle-income countries such as South Africa to include the ECD sector in their planning and budgetary support.

With the social and economic situation manifesting in South Africa through a disproportionate burden of poverty, unemployment, low education levels, and other negative events black South Africans experience [66] this study’s results showed that African children had the highest odds of being stunted compared to children of other races. Similarly, African children are less likely to have medical aid cover and they are also the primary recipients of the CSG because of the legacy of apartheid’s racial segregation and discrimination [54] These results are echoed by Granlund and Hochfeld’s study [9], which ascertains that the child support grant reaches nearly 12 million children and has slightly ameliorated the poverty situation in South Africa. However, despite this intervention, nearly 60% of children still live in households below the upper-bound poverty line [24]. These results suggest that the legacy of apartheid and the political economy has had lasting adverse effects of inequality and food insecurity in households. Furthermore, outbreaks of epidemics such as Ebola and COVID-19 have grave implications for childhood nutrition and threaten children’s survival [68,69]. 

## 5. Limitations of This Study

The study did not measure the quality of care that the child received. Other studies have measured the childcare index by using indexes such as reading to the child, playing with the children, singing songs, and telling stories. In comparison, prior studies have indicated that the quality of parenting/caregiving predicts positive child health outcomes, including child immunisation and breastfeeding. This study did not ask the demographic description of who else cares for the child, such as the age and employment status of the secondary carer. Furthermore, while the study has assessed whether the child was sick in the past few months, it does not consider whether the child received immunisation.

## 6. Conclusions

In conclusion, we observe that stunting and overweight are increasing in South Africa. The income disparities among caregivers resulting from high unemployment rates in the country highlight the importance of socioeconomic status in childcare. The majority of children received multiple forms of care, and many were cared for at home during the day. The private caregiving practices in South Africa are largely unregulated and, thus, require an important policy focus that provides clear direction and caregiving by crèches. The focus of early childhood needs to target children under 5 to create opportunities for them to thrive; hence, there is a need for funding streams by government to support quality childcare in informal settings. We need practical policies to implement innovative strategies to ensure a reversal of the prevailing negative impact of childcare-giving arrangements and children’s malnutrition status in South Africa.

## Figures and Tables

**Table 1 ijerph-20-02572-t001:** Sociodemographic characteristics of mothers of children under five (N = 2966).

Variable	Categories	Frequency	Percentage
Maternal Age	15–24	497	16.74
	25–29	604	20.36
	30–34	602	20.28
	35–39	417	14.06
	40–49	847	28.55
Marital Status	Married/Living with partner	775	26.13
	Single/Divorced	2191	73.87
Religion	No religion	176	5.92
	Christian	2542	85.71
	Other religion	248	8.37
Home language	Afrikaans/English	583	19.67
	Tshivenda/Tsonga	120	4.05
	Nguni	739	24.92
	Sotho	1523	51.35
Children ever born	1 child	1093	36.86
	2 children	845	28.48
	3 children	564	19.02
	4 children	239	8.07
	5+ children	225	7.58
Medical aid	Yes	297	10.02
	No	2669	89.98
Level of education achieved	No schooling	81	2.74
	Primary schooling	337	11.35
	Secondary schooling	1850	62.37
	Tertiary schooling	698	23.54
Employment status	Yes	1252	42.22
	No	1714	57.78
Current activity	Student/Other/Volunteer	207	6.97
	Sick/Home-maker	374	12.61
	Unemployed—active	682	22.98
	Unemployed—discouraged	452	15.23
	Employed	1252	42.22
Emotional health	No depressive symptoms	171	5.77
	Depressive symptoms	2 795	94.23
Water sources	Water carrier/Other	186	6.27
	Public tap	389	13.12
	Piped water at home	2389	80.61
Toilet type	None/Bucket	102	3.44
	Chemical/Pit-latrine	851	28.72
	Flush toilet	2011	67.84
Household electricity	Yes	2682	90.48
	No	282	9.52

**Table 2 ijerph-20-02572-t002:** Child-level characteristics (N = 2966).

Variable	Categories	Frequency (N)	Percentage (%)
Gender	Male	1487	50.14
	Female	1479	49.86
Age in months	0–11 months	306	10.31
	12–23 months	584	19.67
	24–35 months	613	20.68
	36–47 months	698	23.53
	48–59 months	765	25.80
Race	African	2275	76.69
	Coloured	406	13.69
	White/Indian/Asian	285	9.62
Disease episode	No	2609	87.98
	Yes	357	12.02
Sought health care	Yes	233	69.21
	No	103	30.79
Reason for not seeking care	No time/Resources	18	17.67
	Already on treatment	20	20.18
	Child is not sick enough	63	62.15
Childcare during the day	In school	243	8.19
	Crèche/Day mother	730	24.60
	At home	1994	67.22
Receives multiple care	No	980	33.04
	Yes	1986	66.96
Multiple forms of care	Another parent	1070	36.08
	Grandparents	626	21.11
	Relatives/Non-familial	321	10.82
	No other care	949	32.00
Child support grant	No	576	19.41
	Yes	2390	80.59
Medical aid	Yes	205	6.91
	No	2759	93.09

**Table 3 ijerph-20-02572-t003:** Proportion, with percentages, of children cared for during the day and experiencing multiple forms of childcare.

Multiple Forms of Childcare	Childcare during the Day			Total
	At Home	Crèche	In School	N (%)
Another parent	707 (66.04%)	276 (25.81%)	87 (8.15%)	1070 (100%)
Grandparents	409 (65.29%)	162 (25.92%)	55 (8.79%)	626 (100%)
Relatives/non-familial	214 (66.58%)	77 (23.94%)	30 (9.48%)	321 (100%)
No other carer	663 (69.88%)	215 (22.69%)	71 (7.43%)	949 (100%)
Total (N)	1992 (67.16%)	730 (24.61%)	243 (8.19%)	2966 (100%)

**Table 4 ijerph-20-02572-t004:** Individual child and caregiver characteristics associated with child stunting and overweight.

Variables	Categories	Stunting (Model 1)			Overweight (Model 1)		
		Odds	*p*-Value	95% CI	Odds	*p*-Value	95% CI
		Ratio			Ratio		
Gender	Male (RC)						
	Female	0.83	0.036 *	1.01–1.44	1.08	0.483	0.88–1.32
Age of the child	0–11 months	0.88	0.515	0.59–1.30	4.93	0.000 ***	3.16–7.69
	12–23 months	2.32	0.000 ***	1.74–3.08	7.11	0.000 ***	4.84–10.44
	24–35 months	1.77	0.000 ***	1.34–2.34	3.91	0.000 ***	2.66–5.73
	36–37 months	1.40	0.018 *	1.06–1.85	1.74	0.008 **	1.15–2.62
	38–59 months (RC)						
Race	White/Asian (RC)						
	African	0.98	0.874	0.72–1.32	0.98	0.927	0.70–1.39
	Coloured	1.07	0.730	0.74–1.53	0.97	0.875	0.63–1.47
Child ethnic groups	Coloured (RC)						
	English/Afrikaans	1.50	0.149	0.87–2.58	0.99	0.971	0.50–1.97
	Sotho	1.77	0.032 *	1.05–2.98	1.25	0.503	0.65–2.41
	Nguni	1.27	0.360	0.76–2.11	2.23	0.012^*^	1.19–4.20
Child sick 3 times	No (RC)						
	Yes	1.07	0.608	0.82–1.41	0.97	0.866	0.71–1.33
Medical Aid	Yes (RC)						
	No	1.90	0.003 **	1.24–2.93	0.64	0.026^*^	0.43–0.95
Child support grant	No (RC)						
	Yes	1.06	0.648	0.83–1.35	0.91	0.488	0.69–1.19
Multiple care	Grandparent (RC)						
	Parent	1.10	0.428	0.87–1.41	1.04	0.775	0.79–1.38
	Relatives/non-familial	1.68	0.351	0.84–1.62	1.05	0.797	0.72–1.55
	No other care	1.11	0.394	0.87–1.42	1.00	0.992	0.75–1.33
Care during the day	Grade R0/1 (RC)						
	Crèche/Day mother	1.34	0.199	0.86–2.11	1.40	0.266	0.77–2.54
	At home	1.87	0.005 **	1.21–2.88	1.33	0.334	0.75–2.36
Employment status	Unemployed (RC)						
	Employed	1.01	0.910	0.84–1.21	1.11	0.325	0.90–1.38
Marital status	Married (RC)						
	Single/Divorced	0.96	0.671	0.79–1.17	0.92	0.487	0.73–1.16
Toilet type	Flush (RC)						
	None/bucket	0.63	0.063	0.39–1.02	0.91	0.721	0.55–1.52
	Chemical/Pit-latrine	0.94	0.487	0.78–1.12	0.87	0.211	0.70–1.08
Education level	Tertiary education (RC)						
	No education	1.36	0.255	0.80–2.29	1.03	0.935	0.54–1.95
	Primary schooling	1.16	0.357	0.85–1.59	0.95	0.769	0.65–1.38
	Secondary education	0.95	0.612	0.76–1.18	0.93	0.598	0.73–1.20

Reference category (RC), * *p* < 0.05; ** *p* < 0.01; *** *p* < 0.001.

**Table 5 ijerph-20-02572-t005:** Multivariable analysis of risk factors associated with child stunting among children under five.

Variables	Child Stunting (Model 2)			
	Categories	Odds Ratio	*p*-Value	95% CI
Age of the child	0–11 months	0.98	0.954	0.51–1.87
	12–23 months	2.58	0.000 ***	1.60–4.14
	24–35 months	1.72	0.016 *	1.10–2.67
	36–37 months	1.22	0.390	0.77–1.94
	38–59 months (RC)			
Race	White/Asian (RC)			
	African	1.60	0.034 *	1.03–2.47
	Coloured	1.52	0.114	0.90–2.56
Medical aid	Yes (RC)			
	No	2.33	0.003 **	1.35–4.03
Child support grant	No (RC)			
	Yes	1.46	0.030 *	1.04–2.05
Childcare during the day	In school (RC)			
	Crèche/Day mother	1.24	0.531	0.64–2.40
	At home	2.00	0.023 *	1.10-362

Reference category (RC), * *p* < 0.05; ** *p* < 0.01; *** *p* < 0.001.

**Table 6 ijerph-20-02572-t006:** Multivariable analysis of risk factors associated with child overweight among children under five.

Variables	Child Overweight (Model 2)			
	Categories	Odds Ratio	*p*-Value	95% CI
Age of the child	0–11 months	4.81	0.000 ***	2.63–8.81
	1–23 months	8.16	0.000 ***	4.71–14.14
	24–35 months	4.28	0.000 ***	2.48–7.36
	36–37 months	1.57	0.146	0.85–2.95
	38–59 months (RC)			
Childcare during the day	In school (RC)			
	Crèche/Day mother	2.40	0.066	0.94–6.13
	At home	3.82	0.002 **	1.60–9.08

Reference category (RC), ** *p* < 0.01; *** *p* < 0.001.

## Data Availability

The identified data that were used for generating the results in this manuscript can be obtained from National Income Dynamics website (http://www.nids.uct.ac.za/nids-data/data-access accessed on 31 May 2022).

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
