# Peer review of "The Relationship between Childcare-Giving Arrangements and Children’s Malnutrition Status in South Africa"

_ijerph, 2023, doi:10.3390/ijerph20032572_

Round 1
Reviewer 1 Report
The purpose of this study (as perceived) is to reflect on the relationship between the nature of childcare and stunting / overweight status of children under five.
Editorial comments:
Statement on lines 28 and 29 - need reference.
Line 45 - describe the difference between day-care center and a creche.
Rephrase line 68-70.
Reference for NIDS, line 92. Give a detail description of the sample, line 102.
Distinguish between concurrent stunting and overweight, and stunting and / or overweight (line 107). Reference WHO Anthroplus (line 111). Stunting: height-for-age z-score <= -2, and overweight/obese: BMI-for-age z-score >=2 (lines 112,113).
Compare the % cared for at home (69.88%) (line 140) with results reported in abstract.
Rearrange Table 1 - put the third column first, after row heading. Just to verify, does this mean for example, in the first row, if the primary caregiver is the other parent, then this happens in 66.04% of that group, 8.15% of that group are in Gr 1/R, and 25.8% of that group is in creche? Interpret the table - how it should be read. It is extremely important to be clear what exactly is meant by multiple child-care arrangements.
prior research - line 159 - reference?
Line 160 - rather use related than influenced by?
Line 175 - multiple childcare - exactly how determined? categories?
Line 179 - adjusted odds - adjusted for which variables?
Figure 1 - not needed - it gives no additional info.
Table 4: Also, note in the text the categories with higher stunting prevalence. Some results are unexpected such as those with medical aid and receiving grants more stunted?
Line 240-241: results from odds ratios do not correspond with results from Table 4 (with regards to medical aid, for example).
Table 5: was a separate logistic regression applied to each characteristic? confounders? adjusted?
The title "Multiple care" - categories are certain caregivers, does not relate to multiple care? Rather than to who is the caregiver. The concept of multiple care / caregiver should be described in finer detail.
Table 4 and Table 5: Again, check the risk category for medical aid - there might be change in YES and NO. Similarly with race. Stunting has a higher percentage amongst African children, but in Table 5, the risk seems lower (less than 1) in relation to white/Asian children. Please check all variables. Also, exactly which variables were inlcuded in the logistic regression - confounders, etc. Is Table 5 therefore a bivariate logistic regression?
Is Table 6 compiled by adding all those variables as independent variables in one logistic regression function (Multivariate)? How did you decide which variables to include? Race was not significant in Table 5 (ethnic was). Please double check, the OR of African children is now greater than 1, and significant, which makes more sense. Substantiate which variables are included / not included in Table 6. Double check Table 7 similarly.
Line 279 - what is meant by the 40% target?
Line 283 - be specificic about age groups when comparing with other studies - stunting varies substantially amongst different age groups.
Line 290 - reference?
References - redo references such as number 30 and 38 - include additional authors. There are more similar references.
In conclusion: the theme is interesting and this article can make a contribution. Table 1 is actually extremely important in describing multiple care. Unfortunately, when comparing the percentages in Table 1 and Table 3, percentages and concepts do not match. In Table 3, "Receives multiple care" does not feature in Table 1. Similar no match for multiple forms of care (the percentages in Table 3) with Table 1. It all depends from which angle the concept is approached, but it should be described in detail.

Author Response
The a point-by-point response to the reviewer’s comments has been added in the attached file.

Reviewer 2 Report
My comments are attached as a word file. Thank you.

Author Response

(The authors gave the same response as above.)

Reviewer 3 Report
I think this is an excellent study and of great significance to all low incom
Ee countries. It must be read by all policy makers
I think the strength lies in the topic selected adn how important it is to recognise factors beyond diet that cause malnutrition and stunting - Caregiving is so important - in so many low income countries children are left to other caregivers to manage their feeding including older siblings. This topic must be discussed widely for policy change.
It would be useful for the authors to suggest practical options for these mothers to monitor their children's growth so that they can pick up the pause in growth early adn not wait for stunting to occur. It is equally important for them to suggest course corrections for the governments to take note in these situations. In S. Africa, it cannot be ignored if the majority are single parent families. In India, we are seeing a rising trend in stunting so here the role of caregivers and alternate mechanisms need to be spelt out for policy makers to make decisions.
COngratulations to the authors
Author Response
The a point-by-point response to the reviewer’s comment has been added in the attached file.

Round 2
Reviewer 1 Report
Abstract
67.22% were mainly cared for at home – Table 3 indicates 67.16%
….. These results were echoed by multivariable analysis… Which results? Delete this sentence, or say – Some results of multivariate analyses obtained are ……
…Furthermore, in this study 69.88% of, children who were cared for at home by the primary caregiver had no other forms of multiple care arrangements., (for correct interpretation, add “by the primary caregiver).
The link between the mentioned sentence and the next sentence ..This suggests that government needs to look … is not clear.
Introduction
In lines 37-40, edit: weigh-for-height, weight-for-age, height-for-age - see also line 165 and more.
Edit the document, ensure that there is a space before the square brackets indicating reference numbers. Examples, line 40, 49, etc.
Punctuation needs to be improved – lines 50-55 for example. Commas are in wrong places, space before colon, etc. Please edit rest of document as well. See also line 61. 62, 69, 92, 93, 94, 116, 161, 209
Lines 73&74 rephrase – “and having meeting their physical and emotional needs met.”
Lines 86-88: Sentence not complete –“Furthermore, due to the changing dynamics of family structures, characterized by single-parent households where a parent raises a child without a partner living with them.”
Materials and methods
The sample – which area (province? RSA?) were included in the NIDS study? How were the participants selected? More detail is necessary. Was the sample stratified with clusters (enumerator areas maybe) and strata? Was the structure considered in the analyses?
Line 166 – WHO Anthroplus (capital letter?)
Line 167 – children with a z-score of ≤-2 were classified … (minus 2)
Lines 171-172: multiple forms of childcare – different categories of childcare, mainly at home, were described. The reader is confused, in Table 3, it seems as if a child can receive care from someone else (other than a parent, as described here), but additionally, also, even if the carer differs from a parent, at a creche or GradeR/1. Is that what is meant by secondary and multiple care?
The response to this question is still nor clear:
|
Comment: Line 175 - multiple childcare - exactly how determined? categories? |
|
Response: The categories for multiple childcare have been given and the sentence now reads:
The main independent variable (multiple childcare –refers to the care the child receives from others besides the primary care giver and is categorised as the care from the other parent, grandparents, relatives/nonfamily caregivers) |
Line 201 – reference for STATA?
Lines 201-203 – not a complete sentence.
Lines 205-206 – bivariate analysis – Chi-square test – establish the strength of relationships …
No Chi-square tests were performed in this article?
Lines 207-214: The VIF measures the extent of multicollinearity, and identifies variables that should not be included simultaneously as independent variables in a multiple (logistic) regression function. Rephrase the sentences (the VIF is not used to test the strength between the dependent and independent variables). Also, no VIF values were reported in this article?
Lines 212-213 …Variables that depicted a VIF over 4 or a ????????? below 0.25 indicate that 212 multi-collinearity may exist [56].
Lines 213,214: “ At the third level, a binary logistic regression model reporting odds ratios and adjusted odds ratios was generated. “ Binary logistic regression refers to bivariate analysis, adjusted odds ratios can be seen as multivariate analysis.
(distinguish between adjusted logistic regression and multivariate logistic regression - if adjusted, it must be specified which variables are used as confounders, if multivariate analysis is used, specify which variables were included simultaneously in the model).
Table 1
Give total n-value (maybe after the heading Frequency (N=2966). Insert also in Table 2.
Line 231 – rather use concurrent instead of co-existence- check in rest of the document.
Line 233 – “received multiple care (68%)”? The table indicates 67%
Lines 234&235 – the table is incomplete (multiple forms of care).
Line 290 – edit.
Edit lines 293-295:
If you phrase the sentence “xx% of children who were at home during the day….” Then you have to divide 409 by 1992 (20.5%) to establish xx%. If you want to report the 65.29%, it should be written as follows:
…and in this study, 65.29% of children who were cared for by their grandparents, were cared for at home during the day.
69.88% of Children with no other carers besides the primary caregiver were cared for at home during the day.
The interpretation is important. Notice that the rows add up to 100%. This means that “Of the children who were cared for by main caregiver, 69.88% were cared for at home, 22.69% attended a creche and 7.43% are in GradeR/1. Actually, a Chi-square test will be helpful to establish whether there is a relationship between the form of childcare and childcare places during the day.
Does Table 3 imply that even if children are cared for by grandparents of a relative/non family member at home, they still attend a creche or grader/1? Please explain?
Line 322: adjusted binary logistic regression - I understand that this table is only binary logistic regressions – not adjusted.
Lines 325 and further – indicate strength of significance (p<0.05), (p<0.01), (p<0.001) in text and *, **, or *** in Table 4. Include a footnote at Table 4 to describe *,** and ***, and also to explain abbreviations such as (RC)
Footnotes and *,**,*** in tables 5 and 6 as well.
Line 338 – Chi-square test (No Chi-square tests results were reported in this article)
Line 336-338 – which variables were used in the adjusted analyses? Or was this just a multivariate analysis?
|
Comment: Line 179 - adjusted odds - adjusted for which variables? |
|
Response: This sentence has been amended and now reads as follows:
The adjusted odds to account for the other predictor variables (such as age, race, medical aid, child support grant and care during the day) from the multivariate binary logistic regression reported a 95% confidence interval. |
This is still not clear.
In the analyses, Table 5, were the independent variables age of child, race, medical aid, child support grant, childcare during day – all together in one model, OR was the analysis done for age of child, but certain confounders were added such as gender? It seems as if it was the first case, then rather not use the word “adjusted”, and just refer to a multivariate analysis.
Again, it is useful to use the variables that appear significant in the binary logistic regression analysis in the multivariate analysis, as a suggestion. The long explanation in lines 336-338 not needed.
Is there an explanation why receiving a child grant in the binary logistic regression is not significant, but significant in the multivariate logistic regression? Same for race. Also with regards to overweight, childcare during the day (significant in multivariate analysis but not bivariate analysis).
Please look at these comments again:
|
Comment: Table 4 and Table 5: Again, check the risk category for medical aid - there might be change in YES and NO. Similarly with race. Stunting has a higher percentage amongst African children, but in Table 5, the risk seems lower (less than 1) in relation to white/Asian children. Please check all variables. Also, exactly which variables were included in the logistic regression - confounders, etc. Is Table 5 therefore a bivariate logistic regression? |
|
|
|
Comment: Is Table 6 compiled by adding all those variables as independent variables in one logistic regression function (Multivariate)? How did you decide which variables to include? Race was not significant in Table 5 (ethnic was). Please double check, the OR of African children is now greater than 1, and significant, which makes more sense. Substantiate which variables are included / not included in Table 6. Double check Table 7 similarly.
|
|
|
Reviewer 2 Report
Dear Authors,
Thank you for considering the feedback and addressing some of it. The article needs substantive revision for the writing style and presentation of the literature review. Please revise the tables and ensure that all the variable names are consistent, for example, Tables 4 and 5 mention Grade instead of In school. Please follow APA table format to make it very clear, (https://owl.purdue.edu/owl/research_and_citation/apa_style/apa_formatting_and_style_guide/apa_tables_and_figures.html).
None of the model summary statistics were reported in the results. Please consult with a senior writer who has expertise in writing scientific articles.
Please cut the conclusion down to 3 sentences, it is way too long. The introduction and discussion need substantial revision to make them consistent with the research hypotheses.
|
|
